# The Feasibility of an Internal Gas-Assisted Heating Method for Improving the Melt Filling Ability of Polyamide 6 Thermoplastic Composites in a Thin Wall Injection Molding Process

**DOI:** 10.3390/polym13071004

**Published:** 2021-03-24

**Authors:** Thanh Trung Do, Tran Minh The Uyen, Pham Son Minh

**Affiliations:** HCMC University of Technology and Education, Ho Chi Minh City 71307, Vietnam; trungdt@hcmute.edu.vn (T.T.D.); uyentmt@hcmute.edu.vn (T.M.T.U.)

**Keywords:** injection molding, thermoplastic composites, mold heating, mold temperature control, melt filling, thin wall injection molding

## Abstract

In thin wall injection molding, the filling of plastic material into the cavity will be restricted by the frozen layer due to the quick cooling of the hot melt when it contacts with the lower temperature surface of the cavity. This problem is heightened in composite material, which has a higher viscosity than pure plastic. In this paper, to reduce the frozen layer as well as improve the filling ability of polyamide 6 reinforced with 30 wt.% glass fiber (PA6/GF30%) in the thin wall injection molding process, a preheating step with the internal gas heating method was applied to heat the cavity surface to a high temperature, and then, the filling step was commenced. In this study, the filling ability of PA6/GF30% was studied with a melt flow thickness varying from 0.1 to 0.5 mm. To improve the filling ability, the mold temperature control technique was applied. In this study, an internal gas-assisted mold temperature control (In-GMTC) using different levels of mold insert thickness and gas temperatures to achieve rapid mold surface temperature control was established. The heating process was observed using an infrared camera and estimated by the temperature distribution and the heating rate. Then, the In-GMTC was employed to produce a thin product by an injection molding process with the In-GMTC system. The simulation results show that with agas temperature of 300 °C, the cavity surface could be heated under a heating rate that varied from 23.5 to 24.5 °C/s in the first 2 s. Then, the heating rate decreased. After the heating process was completed, the cavity temperature was varied from 83.8 to about 164.5 °C. In-GMTC was also used for the injection molding process with a part thickness that varied from 0.1 to 0.5 mm. The results show that with In-GMTC, the filling ability of composite material clearly increased from 2.8 to 18.6 mm with a flow thickness of 0.1 mm.

## 1. Introduction

In injection molding, the selection of an appropriate cavity surface temperature is a key point in plastic processing, especially with thin products or molding processes with low viscosity materials, such as composites [1,2,3]. With a hot mold surface, the part quality will be improved, although the cycle time will be longer. A lower temperature of the cavity surface will decrease the cooling time, but could lead to an increased number of faults in the product [4,5,6,7]. Therefore, recent research has focused on achieving molding with a high cavity temperature and a cycle time that is as short as possible. In the field of injection molding, thin wall injection molding is used to produce a variety of polymer equipment because of the product price and capability for high-volume production. Most applications of thin wall injection molding are in the field of optical products (such as CDs and DVDs) and micro channel devices. In the field of optics production, the injection molding method also has advantages in the production of waveguides, optical gratings, and optical switches [8,9,10], as well as many products involving micro channel devices, such as capillary analysis systems, micro pumps, and lab-on-a-chip applications [11,12].

In the filling step of the injection molding process, to improve material flow, a high cavity temperature is an important requirement to satisfy the filling of thin wall locations. A high cavity temperature also supports a lower filling pressure. However, if the molding process operates with a high mold temperature, the warpage and shrinkage of parts will occur. In addition, the molding cycle time will be increased for the part that reaches the ejection temperature. To achieve a molding process with a high cavity surface and reduce the frozen layer, many methods of mold temperature control have been investigated in recent years [8]. The initial method used to obtain a high mold temperature was to increase the temperature of cooling water as high as 90 or 100 °C [12,13,14]. When targeting temperatures are higher than 100 °C, heaters are inserted into the mold plate. The thermal energy of the heater thus forms the local heating source of the mold plate [15]. After that, the high temperature from water steam [16] is sometimes used to maintain the temperature of the cavity surface at a high value. However, this method requires that the equipment be operated at high pressure; therefore, the cost of safety equipment is a disadvantage of this method. Further, flexible heating equipment has been suggested and used for auxiliary heating. However, the experimental results show that the method can only increase the cavity temperature by several tens of degrees centigrade.

Instead of heating the entire mold cavity volume, in recent years, many researchers have suggested the use of the mold surface heating method for molding with high cavity temperatures, such as in induction heating [17,18,19,20], high-frequency proximity heating [21,22], and gas-assisted mold temperature control (GMTC) [23,24,25,26,27,28,29,30]. The first two methods support a fast heating rate with a fairly good prediction ability. However, induction heating is applied only for steel molds with a high permeability. On the other hand, besides the advance in fast heating, the induction heating method could easily cause the mold plate to overheat, particularly at the edges. On the contrary, with gas heating, the heating rate is not as high as that of the induction heating, but it can be used for almost the entire mold material. In addition, due to the heat convection between the hot gas and the cavity surface, which has a lower temperature, the gas heating has the ability to prevent the mold from overheating. 

In previous research, the gas heating structure was assembled into the mold to improve the heating efficiency, including the heating rate and temperature distribution [20,21,22,23]. In this design, the hot gas flows and directly exchanges thermal energy to the mold surface. The conduction heating process increases the cavity temperature. Tests of this approach have shown positive results. However, the loss of thermal energy when the air transfers from the heating source to the heating surface is still too great with this approach. This issue is due to the fact that the heating source is separated from the mold or because the heating surface is too far from the heating source. So, to minimize this limitation, the external gas-assisted mold temperature control (Ex-GMTC) method was investigated [24,25,26]. In this molding process, the mold structure is almost the same as the traditional structure. The heating equipment is moved to the heating surface with a robot arm. This method has some advantages as the heating rate and temperature distribution of the cavity surface can be controlled. However, the disadvantage of Ex-GMTC is the cost of additional equipment; furthermore, moving the hot gas generator is not very safe.

Therefore, based on the disadvantages shown in our previous research, in this study, an internal gas-assisted mold temperature control (In-GMTC) approach was used with different levels of insert thickness (t) and gas temperatures to achieve rapid mold surface temperature control for high-aspect-ratio thin wall injection molding. A set of systematic experiments was conducted to correlate the effect of heating conditions, including heating efficiency and temperature distribution uniformity. The feasibility of using internal air-assisted heating for mold surface temperature control during the injection process to improve the melt flow length was evaluated by applying this system to a real molding process with part thicknesses varying from 0.1 to 0.5 mm.

## 2. Simulation and Experimental Method

Internal gas-assisted mold temperature control (In-GMTC) is a new technique in the field of cavity surface heating that can not only provide heating but also facilitate cooling. In general, the goals of mold temperature control are to increase the temperature of the mold surface to the target temperature before filling with the melt and cooling the melt to the ejection temperature. In this research, the internal GMTC (In-GMTC) system consisted of a hot-gas generator system (including an air compressor with an air pressure of 7 bars, an air drier, a gas valve, and a high-efficiency gas heater) and water mold temperature controller, as shown in Figure 1. 

In line with our previous research into the GMTC [22,23], the hot gas generator consisted of an air compressor, an air dryer, a gas valve for volumetric flow control and a high-efficiency gas heater. In this research, the function of the high-power hot gas generator system was to support a heat source, providing a flow of hot air up to 400 °C with an inlet gas pressure of up to 7 bars. In this research, the hot gas generator was hung on the mold so that, in each cycle, the heating process could be started without the support of other equipment such as in the external gas-assisted mold temperature control (Ex-GMTC). Another advantage of this method is that the distance between the hot gas generator and the heating surface was reduced, meaning that the wasted energy was reduced. For the coolant system, a mold temperature controller was used to provide the water at a defined temperature to cool the mold after the filling process and to warm the mold to the initial temperature at the beginning of the experiment. In this research, the initial mold temperature was set at 40 °C. To both control and observe the temperature at the cavity surface, five temperature sensors were used to obtain the real-time mold temperature and to provide feedback to the In-GMTC controller. 

In this research, hot gas was used as a heating source to increase the cavity surface temperature of the injection mold. For the heating operation, first, by opening the mold, two mold plates moved from the closing position to the opening position (Figure 2—Step 1 and Step 2). In Step 2, the supporter moved backward to create a gap between the supporter block and the mold insert. This gap allowed the hot gas to flow and make contact with the mold insert. Second, the hot gas drier was moved to the heating position, as shown in Figure 2—Step 3. Then, the air was heated as it flowed through the gas drier and the outside of the gas drier transferred the hot air, which contacted directly with the cavity surface. This hot gas heated the cavity surface to the target temperature (Figure 2—Step 3). Third, when the mold reached the required temperature, the air supply was turned off (Figure 2—Step 4). Then, the mold was completely closed in preparation for the filling process of the melt (Figure 2—step 5). In Step 5, the supporter block moved toward and contacted with the mold insert so that the mold insert did not become deformed when it stopped due to the filling pressure from the hot melt.

Figure 3 shows the position of the In-GMTC and the mold plate in the injection molding. The gas drier, with a size of 240 mm × 100 mm × 80 mm, is shown in Figure 4. The gas channel was cut inside the gas drier with a width of 5 mm and depth of 10 mm. In this research, the mold cavity was filled by an insert with a size of 77 mm × 77.4 mm. The inserts and the locations of temperature measurement are shown in Figure 5. To observe the heating effect of In-GMTC on the stamp temperature, five temperature measurement points were used. One was located at the top point, which was close to the outlet of the gas drier. The other four were located as shown in Figure 5. In this paper, the influence of stamp thickness on the heating process was observed. To study the temperature distribution of the heating area, a simulation model was built as in the experiment. Because the stamp was inserted into the mold, there was a small air gap between the stamp and the mold; thus, within a short time, this air gap acted as an insulation layer. Therefore, according to previous research [21,22], the simulation model included only two volumes: the stamp volume and the air volume. The geometric view and the meshing model of the system are shown in Figure 6 and Table 1. In this model, the hot gas temperature was varied from 200 to 400 °C under a pressure of 7 bar. The direction of this hot gas flow was set perpendicular to the heating surface. In the simulation, the initial gas volume was set at 40 °C with a pressure of 1 atm. The outlet of the hot gas was set as the opening area, with an air temperature of 40 °C and a pressure of 1 atm. In addition, the initial temperature of the P20 steel insert was set at 40 °C. In order to improve the simulation precision, a hex-dominant element was used to mesh the insert part. To improve the simulation accuracy, a small element size was applied at location S1. In addition, the inflation meshing method was applied with 10 layers at the contact surfaces. In the simulation, the heat transfer mode around all external surfaces of the mold plate was set to free convection to the air, with an ambient temperature of 40 °C and a heat transfer coefficient of 10 W/m^2^ K. The heating process was simulated using ANSYS software (ANSYS, Inc., Ho Chi Minh City, Vietnam) with the same experimental parameters.

To observe the influence of In-GMTC on the melt flow length, the real molding process was performed with the cavity insert as in Figure 5. This insert was added into the cavity plate as shown in Figure 7. This cavity had a size of 46 mm × 8 mm. The insert was manufactured for the experiment with melt flow thicknesses of 0.1, 0.2, 0.3, 0.4, and 0.5 mm. With the common injection molding process, this range of thicknesses represents a kind of thin wall injection molding, which easily results in the short shot problem when the injection pressure is low; however, with an overly high injection pressure, the flash problem easily arises. Thus, with the ability of mold temperature control, the In-GMTC was applied for this molding process to observe the improvement in product quality when the injection molding process was operated with a moderate injection pressure. In this paper, polyamide 6 reinforced with 30 wt.% of glass fiber (PA6/GF30% from Lanxess AG, Cologne, Germany) was used for the molding process, and the molding parameter was maintained for all testing cases, as shown in Table 2. The SW-120B (Shine Well Machinery Co., Ltd., Taichung City, Taiwan) molding machine was used in the experiment.

## 3. Results and Discussions

### 3.1. Effect of Part Thickness on the Mold Temperature Control

In injection molding, the part geometry is an important element that not only impacts on the part formation but also the molding parameters. With a thin product, there are many methods for improving the melt filling by molding at high temperature, such as induction heating and heater heating. Of these methods, gas-assisted mold temperature control has shown many positive results [20,21,22,23]. With gas-assisted mold temperature control, the structure of the stamp insert is often used to increase the heating efficiency. Based on the results of these research works, the stamp thickness is one of the most important parameters of mold design and is impacted by the part thickness. Therefore, in this paper, to estimate the heating ability of In-GMTC, an insert with a size of 77.4 mm × 77 mm was inserted into the cavity plate, and the heating process was achieved with a hot gas temperature of 300 °C; the gap between the gas gate and the heating surface was 3.5 mm (Figure 5c). By simulation with the model, as shown in Figure 6, the results of the variation in the mold temperature (at sensor S1—Figure 5) versus time for a heating time of 20 s are described as shown in Figure 8. For an initial mold temperature of 40 °C, it can be seen that the In-GMTC heated the plate to above 170.4 °C. This shows that the In-GMTC can support a heating rate of 6.5 °C/s. With the change in part thickness from 0.1 to 5.0 mm, the temperature at sensor S1 varied from 164.3 to 170.4 °C. This temperature range was higher than the glass transition temperature of almost all common plastic material. In former papers on gas-assisted mold temperature control [20,21,22], when the GMTC was applied for a heating area of 58 mm × 30 mm, a gas flow rate of 500 L/min and a gas temperature of 300 °C, the maximum heating rate was only about 2.2 °C/s [25]. This means that the In-GMTC, with the system design and heating process shown in Figure 2, has a great advantage in terms of the heating efficiency of the mold surface. 

To observe the influence of part thickness on the heating step, five stamp thicknesses were used for the experiment. Figure 8 shows that with thicknesses of 0.1, 0.2, 0.3, 0.4, and 0.5 mm, the temperature at sensor S1 varied from 164.3 to 170.4 °C; this means that the heating rate increased from 6.2 to 6.5 °C/s. The increase in heating rate when the part thickness increased can be explained by the thermal energy needed to heat the stamp volume. Because the stamp and the mold plate were separated by an isolation layer, with the same heating source, the heating result mainly depended on the stamp volume. So, with a thinner stamp, more thermal energy is needed to increase the stamp temperature. Based on the design shown in Figure 5c, because the stamp thickness and gap between the gas gate to the heating surface were 5 and 3.5 mm, respectively, with a thicker part, the less material there was at the cavity; therefore, a better heating rate was achieved with a thicker product. However, the difference in the heating rate when the part thickness changed from 0.1 to 0.5 mm was small. Therefore, in thin wall injection molding, this property is an advantage of this heating method, which could support mold temperature control with a thin wall part and a thickness that could be varied to lower than 0.5 mm.

The temperature curves in Figure 8 show that the heating rate was extremely high in the first 2 s, with the heating rate varying between 23.5 and 24.5 °C/s. This heating rate is higher than those of many heating methods reported in recent years [19,20,21,22,23,24,25]. After the first 2 s, the heating rate decreased. Although the temperature of the heating surface was not the same as the cavity surface (which was measured) as in Figure 5c, the temperature change was almost the same as those shown in other research on gas-assisted mold temperature control, which can be explained by the absorption of thermal energy and heat transfer. In this research, the heating surface absorbed the thermal energy, and the thickness at sensor S1 was thin, meaning that this thermal energy was held within a small material volume, resulting in a rapid temperature increase at sensor S1. This phenomenon was clearly apparent in the first 2 s. However, the higher temperature meant that the thermal energy at the thin cavity location transferred to another area with a lower temperature. So, this phenomenon lowered the heating rate at the cavity surface. In addition, with a heating time of 20 s, the temperature curves did not show the same limitation as in other studies [24,25]. Therefore, the temperature at the cavity surface still increased with a longer heating time or a high-powered heating source. However, as mentioned above, the temperature at 20 s was high enough to facilitate the melt flowing, and so based on this simulation, the heating time of 20 s was used for the following cases.

In our former study, when the internal GMTC was used for mold surface heating, there was a temperature difference between the inlet and outlet area [20,21,22]. Therefore, in this research, to evaluate the uniformity of the heating process under various stamp thicknesses, the temperature distribution at the cavity surface of the stamp, as shown in Figure 5a, was measured and compared using a simulation and experiments. Figure 9 shows the simulation result regarding the temperature distribution of the cavity. The temperatures of five sensors were measured at the end of the heating step and are compared in Figure 10. These results show that the highest temperature was located at the top of the stamp (sensor S1), which was closest to the hot gas gate, and the temperature was lower at the bottom of the stamp. This kind of distribution was better than that of previous research on the internal GMTC, which often found unbalanced temperatures between the two sides of the cavity area [20,22]. In addition, compared with the induction heating method [14,16,17], the In-GMTC method solved the problem regarding the low temperature at the center of the heating area and would therefore be better for application in real molding products. This result also shows that the temperature differences between the five sensor locations were 81.3, 81.1, 80.5, 79.1, and 78.2 °C with product thicknesses of 0.1, 0.2, 0.3, 0.4, and 0.5 mm, respectively. These results also prove that, with a heating time of 20 s, for all types of stamp thickness, the temperature of the cavity varied from over 83.8 to approximately 164.5 °C and the higher temperature was close to the melt entrance, which could lead to a greater reduction in the frozen layer. Thus, the pressure drop of the hot melt was limited and the melt could flow faster. This means that this temperature distribution was suitable for use in the injection molding process. In addition, these results also show that the lowest temperature could almost satisfy the mold temperature of common plastic materials and the highest temperature was not so high that degradation of plastic materials would occur. 

To verify the accuracy of the simulation result, the experiment achieved the same boundary condition as that in the simulation. The experiment was performed 10 times for each case; after that, the average value was represented for each case. Then, the temperature at the sensors was measured and compared with the simulation result, as shown in Figure 10. The comparison shows that the temperature difference between the simulation and experiment was lower than 10 °C. This difference is due to the measurement delay of the sensor, especially as, in this state, the thermal energy transferred quickly from the higher temperature area to the lower temperature area. However, in general, this result shows that the results of the simulation and the experiment have a good agreement.

### 3.2. Effect of the Inlet Temperature on the Heating Process

In the GMTC method, the gas temperature is an important element, which can be represented by the heating source. In the application of GMTC, a higher gas temperature causes the heating rate to increase. However, the amount of wasted energy will also increase. Therefore, in this study, this was investigated with the model shown in Figure 5 and Figure 6 with gas temperatures of 200, 300, and 400 °C, a product thickness of 0.5 mm corresponding to the stamp thickness of 1 mm and a heating time of 20 s.

Figure 11 and Figure 12 show the temperature distribution under different air temperatures with a stamp thickness of 1 mm. This result shows that with the higher inlet temperature, the heating process became more effective, resulting in a higher temperature at the center of the plate, as well as the temperature difference on the plate increasing. In detail, based on the simulation result, the temperature differences along the center line of the stamp were 54.2, 79.1, and 85 °C, with inlet temperatures of 200, 300, and 400 °C, respectively. In this paper, a comparison was performed between three different inlet temperatures, as shown in Figure 13a. To verify the accuracy of the simulation result, the experiment was performed with the same gas gap. The temperature at five sensors was collected and compared, as shown in Figure 13b–d, and the temperature distribution is shown in Figure 12. The comparation shows that the simulation and the experiment have a good agreement. In other research on the GMTC [20,21,22,23,24,25], the heating process was influenced by another wild source; however, with the In-GMTC, the heating was achieved in the private volume, meaning that the simulation and the experimental result exhibited good agreement. Therefore, this heating method could be easier to predict by simulation than the external GMTC. Figure 13b–d shows that the mean temperature in the experiment was slightly lower than the simulation result. This was due to the fact that the simulation results show the temperature at the end of heating step exactly; however, in the experiment, there was a delay time associated with the thermal camera obtaining the thermal picture, and in this delay time, the thermal energy at the higher temperature area was transferred to the lower temperature area, resulting in a lower temperature being obtained by the thermal camera.

### 3.3. Improve the Melt Flow Length of the Polyamide 6 Thermoplastic Composites by Internal Gas Heating for the Gate Temperature Control

To verify the efficiency of In-GMTC for use in the mold temperature control, the mold of melt flow length testing was used for experiment. The dimension of cavity is shown in Figure 5a. The melt flow thickness varied from 0.1 to 0.5 mm. The injection molding experiment was carried out with PA6/GF30% plastic and the molding parameters are shown in Table 3. For the common injection molding cycle, the mold temperature should be set in the range of 20–80 °C; however, with the thin wall product as in this case, to fill the cavity, the mold temperature must be set as high as the system can tolerate. This set up allows for easy flow due to the reduction in the freeze layer of the melt flow [3]. However, when the mold temperature is high, energy is inevitably wasted, and other problems occur such as warpage or flashing. To avoid these problems, local mold temperature control is presented in this paper. Instead of maintaining the entirety of the mold plate at the high temperature, local mold temperature control was achieved for the cavity area by applying local air pre-heating at the beginning of the molding cycle. The high temperature at the gate area reduced the pressure drop of the melt flow when it passed the area [10]. Figure 7 shows the cavity plate with an insert, which included the cavity area and the gate area. In the same manner as the above structure, the gate area was re-designed with a steel insert to improve the heating efficiency. This insert had a dimension of 77 mm × 77.4 mm × 5 mm. To observe the influence of gas temperature on the heating process, gas with temperatures of 200, 250, 300, 350, and 400 °C were used with a heating time of 20 s. To verify the heating efficiency as well as the ability to perform local heating, an infrared camera was used to obtain the temperature distribution at the end of the heating step. After that, the real molding cycle was achieved with the parameters shown in Table 3. For each gas temperature, the molding cycle was operated for 20 cycles to stabilize all of the systems; then, the product of the next 10 cycles was collected to compare the melt flow lengths. Figure 14 shows the temperature distribution and the melt flow pattern with different gas temperatures. The gate temperature and melt flow length were measured. They are shown in Table 3. The temperature distribution shows that the high temperature was focused only at the gate area, which was heated by the hot gas for 20 s. Therefore, the mold plate was maintained at the low temperature in all molding cycles, which reduced the warpage and flashing, as well as the amount of energy wasted when compared with the common case. In the case without heating, the melt flow pattern shows that the melt length was 17 mm when the gate temperature was only about 50 °C. With the In-GMTC, when the gas temperature increased from 200 to 400 °C, the gate temperature varied from 50 to 216 °C with a heating time of 20 s. In other research, when the mold temperature was higher than the glass transition temperature, the melt flowed easily [16,17,18,19,20,21,22]. Thus, in this paper, Figure 14 shows that when the gas temperature was higher than 350 °C, the cavity was completely filled. In addition, these results for a 0.5 mm part thickness also prove that the In-GMTC leads to a large improvement in the melt flow length for the thin wall injection molding product—an increased melt flow length from 36.9% to 100% (fully filled) was observed when the mold plate was maintained at approximately a common temperature. The filling ability of PA6/GF30% for other part thicknesses was compared, as shown in Figure 15. In general, these results show that the In-GMTC method has a positive influence on the flow ability of PA6/30%GF, which was verified with flow with part thicknesses of 0.1–0.5 mm.

## 4. Conclusions

In this study, an internal gas-assisted mold temperature control (In-GMTC) under different flow thicknesses (t) (0.5, 1.0, 1.5, and 2.0 mm) and a gas temperature that varied from 200 to 400 °C was applied to achieve rapid mold surface temperature control. Then, the In-GMTC was applied to verify the melt flow length of PA6/GF30% material in the thin wall injection molding cycle. Based on the results, the following conclusions were obtained:The influence of the insert thickness on the heating process was unclear. For a small heating area, a thinner stamp provided a higher heating rate. The heating rate was about 6.5 °C/s over 20 s. However, for the first 2 s, an extremely high heating rate was observed—24.5 °C/s.In combination with the insert thickness, the gas temperature is an important parameter, as it has a large impact on the heating rate, as well as the temperature difference at the cavity surface. The temperature differences along the center line of the stamp were 54.2, 79.1, and 85 °C with inlet temperatures of 200, 300, and 400 °C, respectively.The results for the temperature distribution show that the highest temperature was located at the top of stamp (sensor S1), which was closest to the hot gas gate, and the temperature was lower at the bottom of the stamp. This kind of distribution is more favorable than those of previous research on the internal GMTC, which often obtained unbalanced temperatures between two sides of the cavity area.Using the ANSYS software with the CFX module, the heating process by In-GMTC could be predicted with good accuracy. The comparison between the simulation and experiment proved that because the heating method in this paper was used in the private volume, the simulation and the experimental results show good agreement.The application of In-GMTC for the real molding cycle shows that the melt flow length was clearly improved when the In-GMTC was used at the back of the mold insert, which increased the melt flow length from 36.9% to 100% (fully filled) when the mold plate was maintained at approximately a common temperature. The filling ability of PA6/GF30% with other part thicknesses was explored, and the results show that the In-GMTC method has a positive influence on the flow ability of PA6/GF30%, which was verified for part thicknesses of 0.1–0.5 mm.

## Figures and Tables

**Figure 1 polymers-13-01004-f001:**
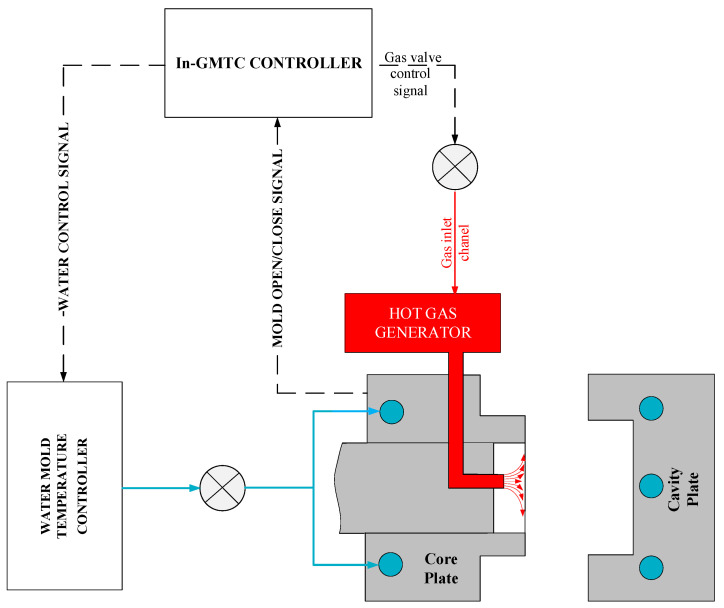
The internal gas-assisted mold temperature control (In-GMTC) system.

**Figure 2 polymers-13-01004-f002:**
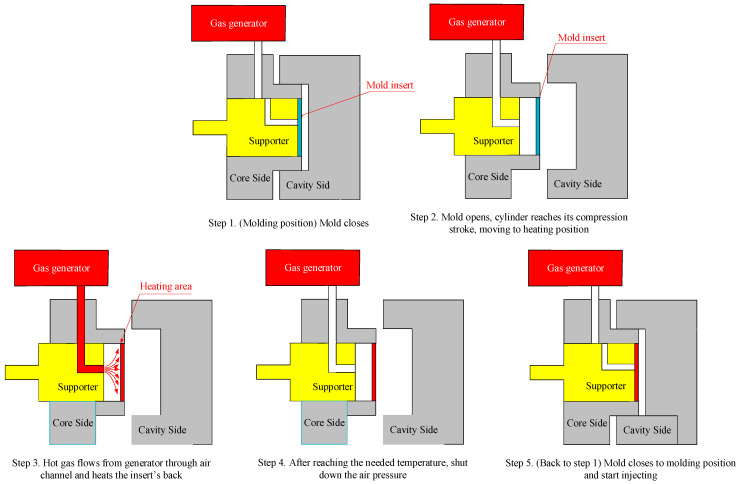
Mold position in the heating stage of the In-GMTC process.

**Figure 3 polymers-13-01004-f003:**
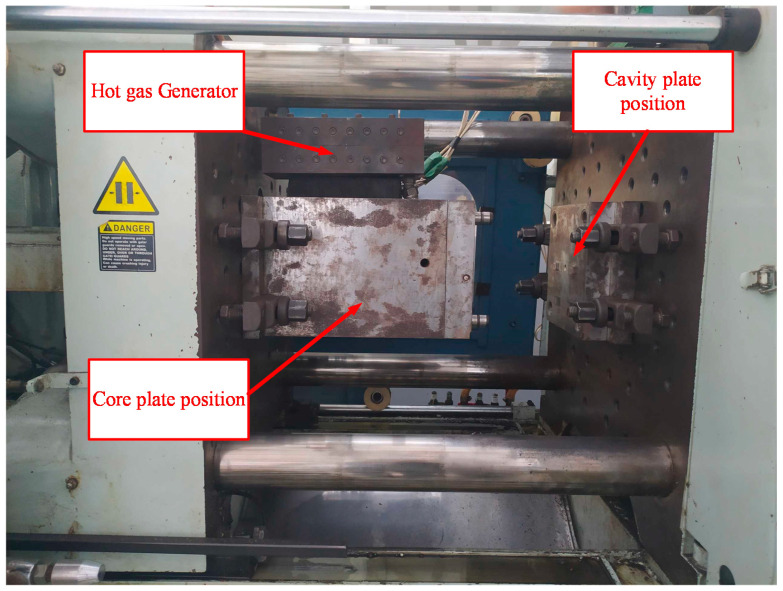
The experimental setup for In-GMTC.

**Figure 4 polymers-13-01004-f004:**
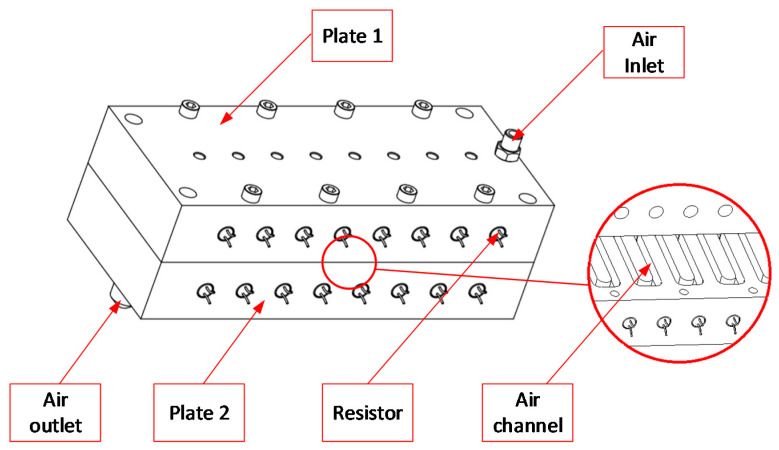
The hot gas generator.

**Figure 5 polymers-13-01004-f005:**
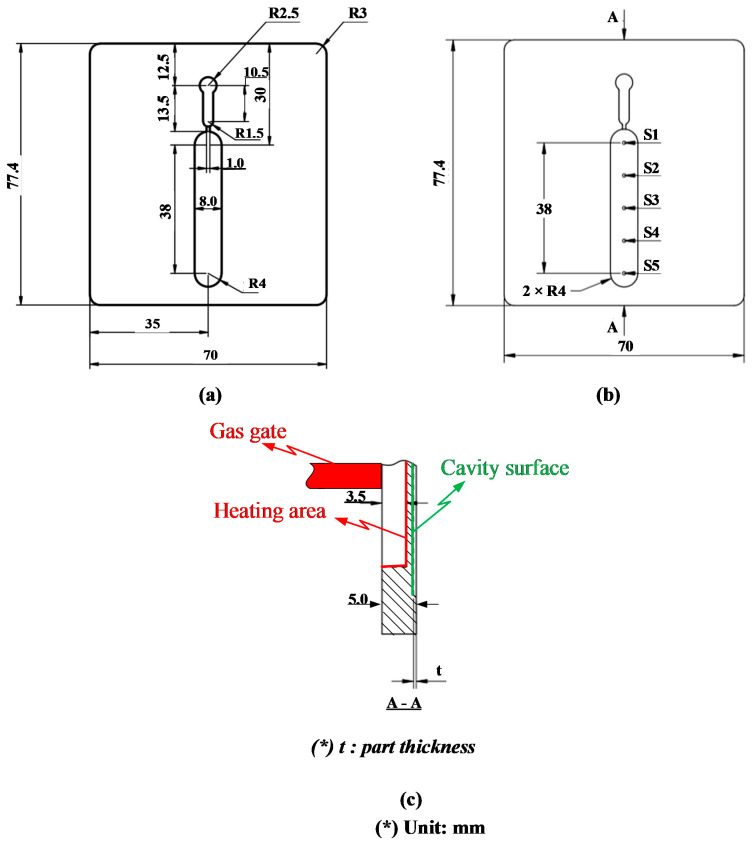
Insert dimension (**a**) with the sensor locations S1 to S5 (**b**) and the heating surface of the insert (**c**).

**Figure 6 polymers-13-01004-f006:**
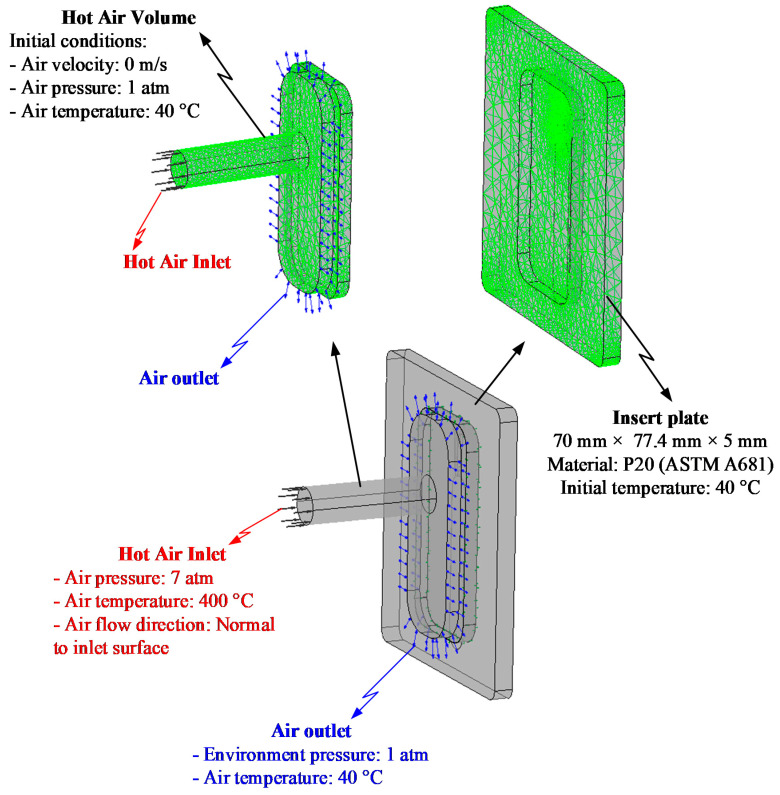
Boundary conditions and the meshing model at the insert area.

**Figure 7 polymers-13-01004-f007:**
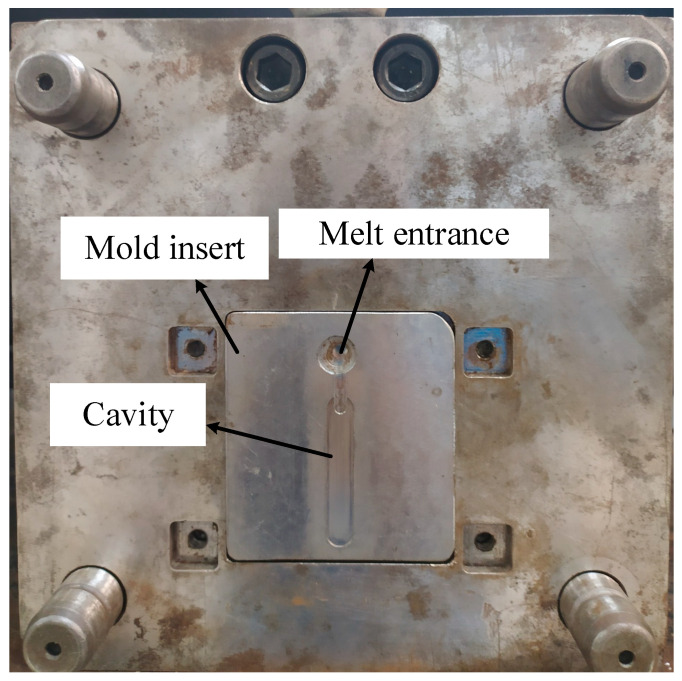
The mold for the melt flow length testing with the gate heating area.

**Figure 8 polymers-13-01004-f008:**
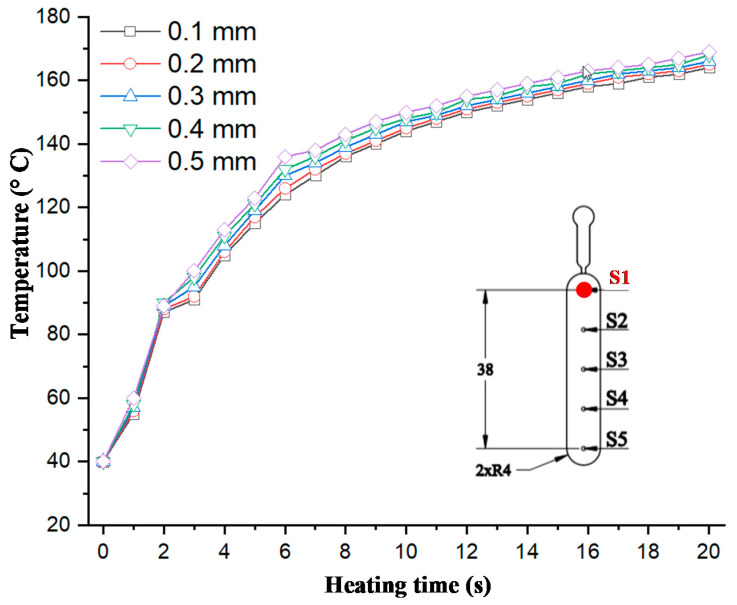
Temperature comparison at the center of the heating area (Point S1) in simulations with different product thicknesses.

**Figure 9 polymers-13-01004-f009:**
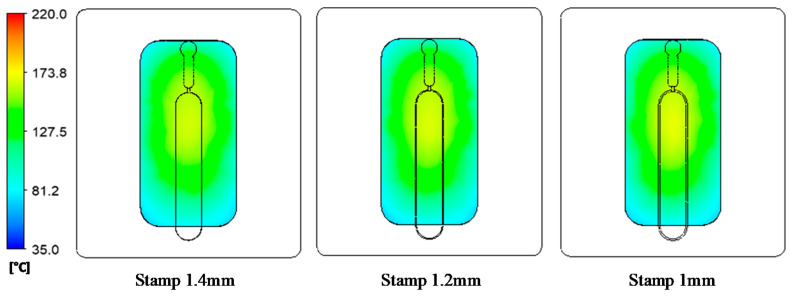
Temperature distribution of the molding area with an initial temperature of 40 °C, a gas temperature of 300 °C, an inlet gas pressure of 7 bars, and a heating time of 20 s.

**Figure 10 polymers-13-01004-f010:**
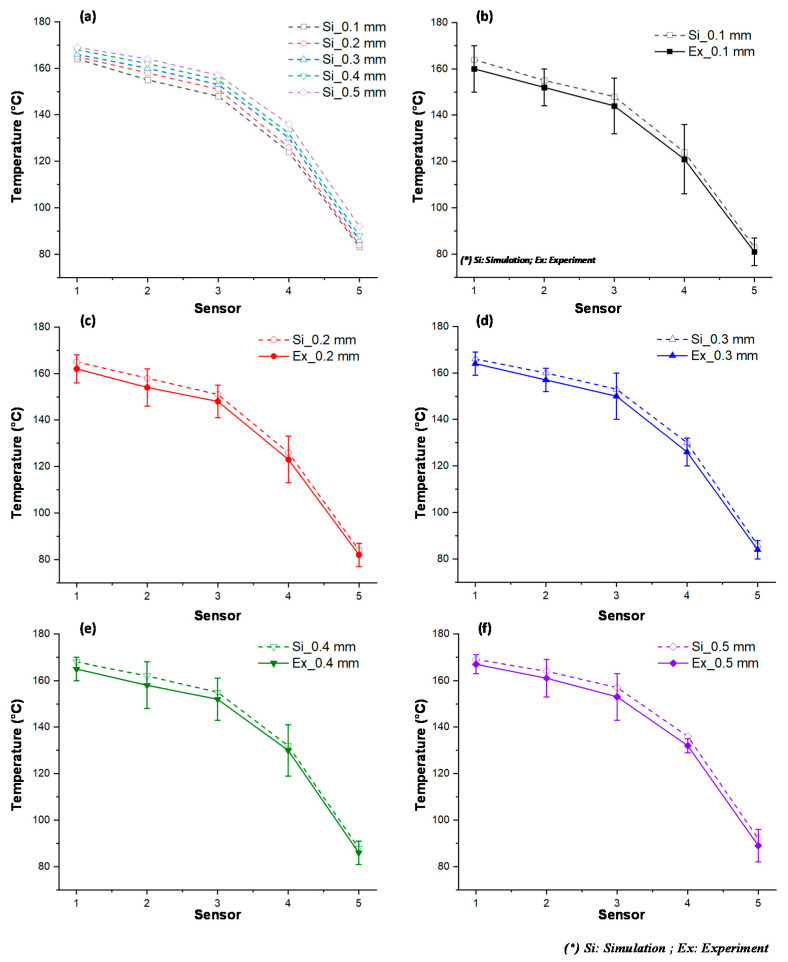
Temperature along the center line in simulations with different product thicknesses.

**Figure 11 polymers-13-01004-f011:**
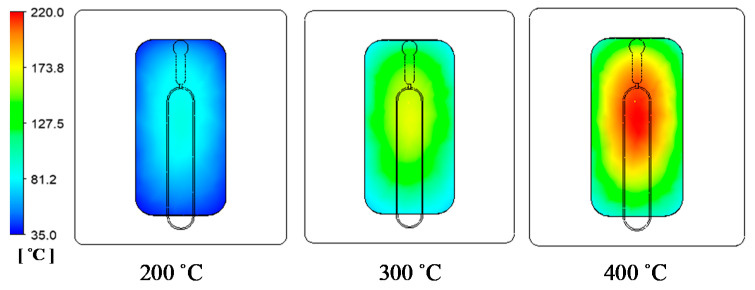
Temperature distribution of the stamp under different air inlet temperatures with a heating time of 20 s.

**Figure 12 polymers-13-01004-f012:**
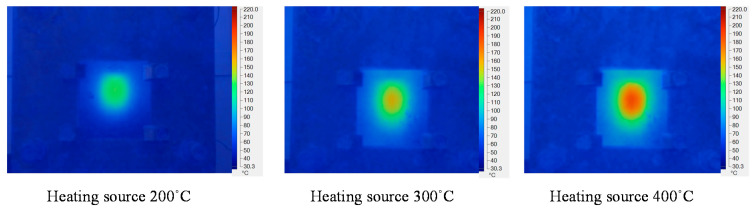
Temperature distribution of the cavity plate after 20 s of heating under different hot gas temperatures.

**Figure 13 polymers-13-01004-f013:**
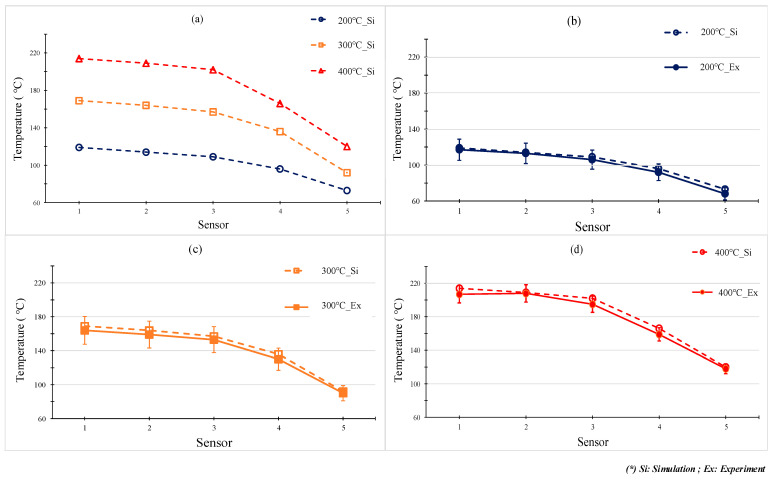
The comparison of temperatures along the center line by simulation (**a**) with different inlet temperatures with a heating time of 20 s and stamp thickness of 0.5 mm under gas temperature of 200 °C (**b**); 300 °C (**c**) and 400 °C (**d**)**.**

**Figure 14 polymers-13-01004-f014:**
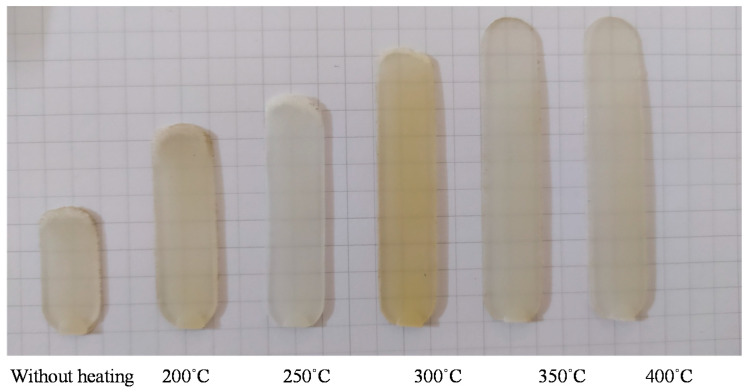
The melt flow length of a 0.5 mm part thickness with and without In-GMTC under different hot gas temperatures.

**Figure 15 polymers-13-01004-f015:**
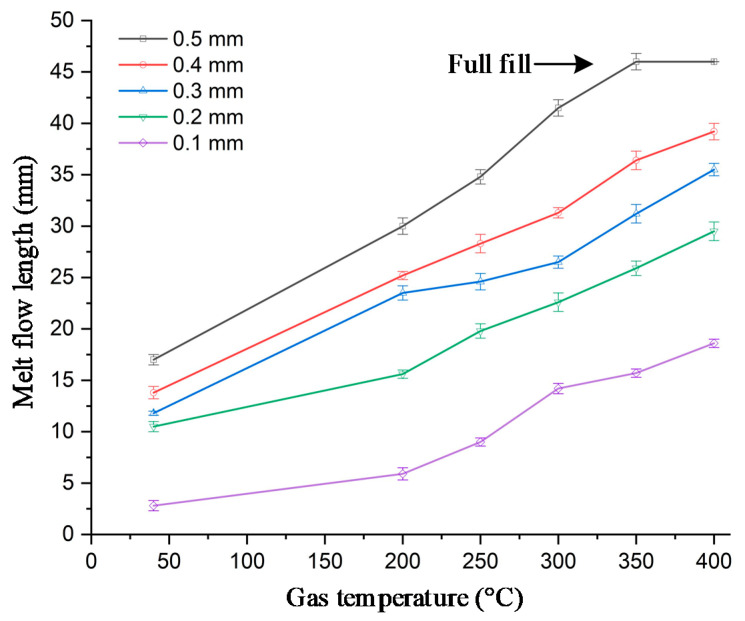
The melt flow length and gate temperature under different gas temperatures for In-GMTC.

**Table 1 polymers-13-01004-t001:** Material properties (for simulation).

Material	Properties	Unit	Value
Air	Molecular mass	kg/kmol	28.96
	Density	kg/m^3^	1.185
	Specific heat capacity	J/kg K	1004.4
	Dynamic viscosity	kg/ms	1.831 × 10^−5^
	Thermal conductivity	W/mK	0.0261
Steel	Molecular mass	kg/kmol	55.85
	Density	kg/m^3^	7854
	Specific heat capacity	J/kg K	434
	Thermal conductivity	W/mK	60.5

**Table 2 polymers-13-01004-t002:** The molding parameters for the product of the front cover plate.

Molding Parameter	Unit	Value
Injection speed	cm^3^/s	23
Injection pressure	atm	35
Injection time	s	1.5
Packing time	s	2
Packing pressure	atm	35
Cooling time	s	15
Mold temperature	°C	50
Melt temperature	°C	265
Pre-heating time by In-GMTC	s	20

**Table 3 polymers-13-01004-t003:** The experimental results of an internal gas-assisted mold temperature control (In-GMTC) for the melt flow length of polyamide 6 reinforced with 30 wt.% glass fiber (PA6/GF30%).

Heating Results.	Part Thickness	Hot Gas Temperature (°C)					
		Without Heating	200	250	300	350	400
Melt flow length (mm) (percent filling)	0.5 mm	17 (36.9%)	30 (65.2%)	34.8 (75.6%)	41.5 (90.2%)	46 (100%)	46 (100%)
	0.4 mm	13.8 (30.0%)	25.2 (54.8%)	28.3 (61.5%)	31.3 (68.0%)	36.4 (79.1%)	39.2 (85.2%)
	0.3 mm	11.8 (25.7%)	23.5 (51.1%)	24.6 (53.5%)	26.5 (57.6%)	31.2 (67.8%)	35.5 (77.2%)
	0.2 mm	10.5 (22.8%)	15.6 (33.9%)	19.8 (43.0%)	22.6 (49.1%)	25.9 (56.3%)	29.5 (64.1%)
	0.1 mm	2.8 (6.1%)	5.9 (12.8%)	9.0 (19.6%)	14.2 (30.9%)	15.7 (34.1%)	18.6 (40.4%)

## Data Availability

The data used to support the findings of this study are available from the corresponding author upon request.

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
