# Peer review of "The Feasibility of an Internal Gas-Assisted Heating Method for Improving the Melt Filling Ability of Polyamide 6 Thermoplastic Composites in a Thin Wall Injection Molding Process"

_polymers, 2021, doi:10.3390/polym13071004_

Round 1

Reviewer 1 Report

High mold temperature can really reduce the flow resistance and extend the flow length. The authors proposed an internal gas-assisted mold temperature control to preheat the mold surface and study the gas temperature effects on the composite polymer melt flow length. It is an innovative design which can gave a glimmer of hope for injection molding in the future. I think it is worth to be published in the journal.

The authors conducted simulations for their experiments and provided part temperature distributions. However, I concern the mold temperature distribution after preheating by hot gas because it relates the cooling system design and cooling time. If the full cycle temperature change of the mold is provided it can be more useful for the plastic manufactures.  

Author Response

Dear Reviewer,

Thank you for your letter and constructive comments concerning our manuscript entitled “The Feasibility of an Internal Gas-Assisted Heating Method for Improving the Melt Filling Ability of Polyamide 6 Thermoplastic Composites in a Thin Wall Injection Molding Process”. We have studied your comments carefully and think about your idea of cooling step as:

The authors conducted simulations for their experiments and provided part temperature distributions. However, I concern the mold temperature distribution after preheating by hot gas because it relates the cooling system design and cooling time. If the full cycle temperature change of the mold is provided it can be more useful for the plastic manufactures

--> Response:

Thank you very much for your mention about the cooling step, which is an important step in injection molding process. Actually, in this manuscript, we just focus on the heating step. However, in future, the cooling step will be researched with the model as in this figure:

(please read the Figure in attached file)

In this design, the gap between the support block and the insert will provide a space for heating and cooling the insert. Therefore, in this design, the cavity surface could be heated as in this manuscript and cooled faster than the case of cooling channel due to the distance between the water and the cavity surface is very thin.

We hope that this idea will be got positive result in experiment and as soon as possible, we could show out the result to other researchers.

We sincerely hope that these explanation meet your approval.

Thank you very much for your advices.

Yours Sincerely,

PHAM SON MINH

HCMC University of Technology and Education, Hochiminh city, Vietnam

No 1 Vo Van Ngan Street, Linh Chieu Ward, Thu Duc District, Ho Chi Minh City, Vietnam

Room E1.107

Phone No: +84-(938)-226313

E-mail: minhps@hcmute.edu.vn  

Reviewer 2 Report

The authors present a very readable work related to the gas assisted heating to improve the melt filling. I think the whole work has a very engineering point of view, but is still fluent and pleasurable.

I have a major concerning related to the experimental part. I would highlight the substances you have used and the machinery you employ: the data you give in line 194 and further I would really prefer to have them in a separate section, in order to catch the reader attention and to help people who want to repeat your job.

I have some minor revisions as follow:

Figure 5: please correct and the graphic dimension that at 100% zoom result not readable.

Table 1 and whole document: degree K does not require the º symbol but only the K letter, please correct them.

Figure 10: the little molding as inset of all the graphics are totally not visible, either you delete all of them or you enlarge.

Figure 9&11: down to the degree scale bar it is missing the º symbol.

I have tick inappropriate self-citation because I personally avoid to pass the 5 self-citations. In this document I have red 7 citation related to Pham Son Minh, which result too much in my opinion.

Author Response

Dear Reviewer,

Thank you for your letter and constructive comments concerning our manuscript entitled “The Feasibility of an Internal Gas-Assisted Heating Method for Improving the Melt Filling Ability of Polyamide 6 Thermoplastic Composites in a Thin Wall Injection
Molding Process
”. We have studied your comments carefully and made revisions that we hope meet will with your approval. The modification was highlighted with the note in the manuscript. Here are our responses (your comments are in highlighted bold italics):

  1. I have a major concerning related to the experimental part. I would highlight the substances you have used and the machinery you employ: the data you give in line 194 and further I would really prefer to have them in a separate section, in order to catch the reader attention and to help people who want to repeat your job.

-->Response 1:

The data of plastic material and injection machine will be followed the online paper. On the other hand, if the reader needs these data, they can contact with the Correspondence author by e-mail, we will send these data to the reader.

Another thing is about the source of plastic material, we already checked again, in this research, the PA6 30%GF was supported by the from Lanxess AG, Cologne, Germany, so, we already modified this information in the manuscript at line 195.

  1. Figure 5: please correct and the graphic dimension that at 100% zoom result not readable.

--> Response 2:

The layout and the size of dimension of Figure 5 was modified for clearer as in line 180.

  1. Table 1 and whole document: degree K does not require the º symbol but only the K letter, please correct them.

à Response 3:

Thank you very much for your precise comment. The “degree K was corrected as “only K letter” (line 184)

  1. Figure 10: the little molding as inset of all the graphics are totally not visible, either you delete all of them or you enlarge?

--> Response 4:

Thank you for your suggestion, the picture of insert in Figure 10 was deleted (line 306-307).

  1. Figure 9&11: down to the degree scale bar it is missing the º symbol.

--> Response 5:

The degree symbol (°) was added into the unit of scale bar (line 294, 340)

  1. I have tick inappropriate self-citation because I personally avoid to pass the 5 self-citations. In this document I have red 7 citation related to Pham Son Minh, which result too much in my opinion.

--> Response 6:

Thank you very much for your careful in the self-citation of our manuscript, so, after check again, the modification was made as: Reference 22 (line 501) and 26 (line 512) were changed to the papers from other researchers.

We sincerely hope that these modification meet your approval.

Thank you very much for your advices.

Yours Sincerely,

PHAM SON MINH

HCMC University of Technology and Education, Hochiminh city, Vietnam

No 1 Vo Van Ngan Street, Linh Chieu Ward, Thu Duc District, Ho Chi Minh City, Vietnam

Room E1.202

Phone No: +84-(938)-226313

E-mail: minhps@hcmute.edu.vn  
